# Sparse Probabilistic Graph Circuits

**Martin Rektoris**[1]     **Milan Papež**[1]     **Václav Šmídl**[1]     **Tomáš Pevný**[1]

[1]Artificial Intelligence Center, Czech Technical University, Prague, Czech Republic

## Abstract

Deep generative models (DGMs) for graphs achieve impressively high expressive power thanks to very efficient and scalable neural networks. However, these networks contain non-linearities that prevent analytical computation of many standard probabilistic inference queries, i.e., these DGMs are considered *intractable*. While recently proposed Probabilistic Graph Circuits (PGCs) address this issue by enabling *tractable* probabilistic inference, they operate on dense graph representations with $\mathcal{O}(n^2)$ complexity for graphs with $n$ nodes and $m$ *edges*. To address this scalability issue, we introduce Sparse PGCs, a new class of tractable generative models that operate directly on sparse graph representation, reducing the complexity to $\mathcal{O}(n+m)$, which is particularly beneficial for $m \ll n^2$. In the context of de novo drug design, we empirically demonstrate that SPGCs retain exact inference capabilities, improve memory efficiency and inference speed, and match the performance of intractable DGMs in key metrics.

## 1 INTRODUCTION

Deep generative models (DGMs) for graphs have gained considerable attention due to their broad applicability in various fields, including chemistry [De Cao and Kipf, 2018], biomedicine [Ingraham et al., 2019], cybersecurity, and social network analysis [Wang et al., 2024]. DGMs represent probability distributions, so it is natural to expect that they should support basic probabilistic inference tasks. However, this is typically not the case, because most DGMs [Vignac et al., 2023, Jo et al., 2022, Shi et al., 2020, Liu et al., 2021] are implemented using highly non-linear deep neural networks. The key reason is that standard inference queries (such as marginalization, conditioning, and expectation)

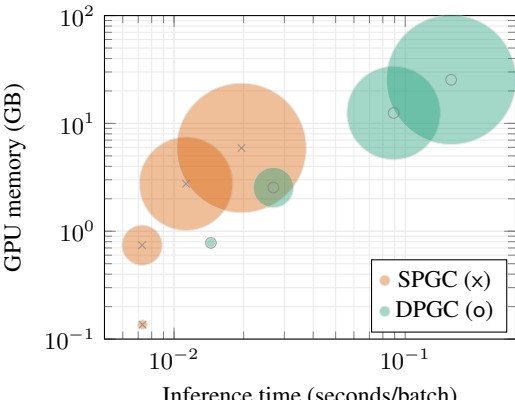

Figure 1: The inference time and peak memory consumption of a single batch with 256 instances for Sparse PGCs (SPGCs) and Dense PGC (DPGCs). The circle size corresponds to the maximum number of nodes, $n_{\max}$, of different datasets, ranging from the smallest to the largest: QM9, Zinc250k, Guacamol, and Polymer.

require evaluating integrals over the model's distribution. The presence of non-linearities in neural networks prevents us from finding efficient, closed-form solutions to most integrals of interest. To overcome this issue, researchers often rely on expensive numerical approximations or query-specific, non-universal solutions. In contrast, tractable generative models offer exact and efficient computation for a broad class of inference queries [Vergari et al., 2021]. These capabilities have proven useful in other domains for tasks like missing value imputation, explainability [Peharz et al., 2020a, Choi et al., 2020], and uncertainty quantification, but remain rather unexplored for graph-structured data.

To bridge this gap, recent work introduced Probabilistic Graph Circuits (PGCs) [Papež et al., 2025], a class of tractable deep generative models for graphs. PGCs extend the framework of Probabilistic Circuits (PCs) [Choi et al., 2020] for graph-structured data, building upon earlier work on GraphSPNs [Papež et al., 2024a] and SPSNs [Papež et al., 2024b]. However, the existing version of PGCs relies on a dense graph representation. For a graph with $n$ nodes, this leads to modeling a full $n \times n$ adjacency matrix, which re-

*Accepted for the 8th Workshop on Tractable Probabilistic Modeling at UAI  (TPM 2025).*

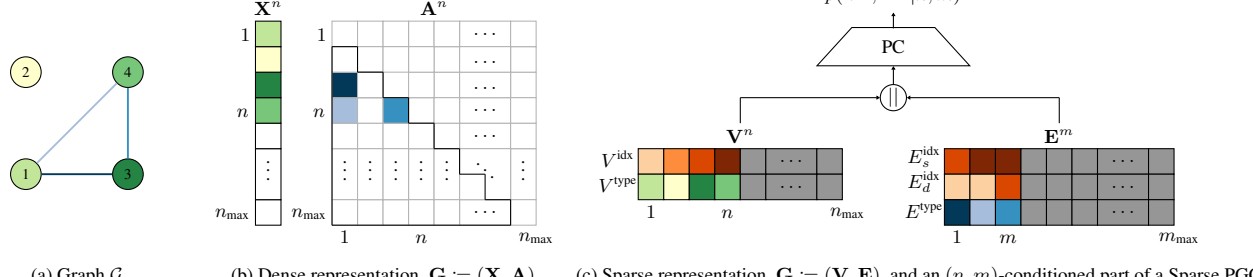

(a) Graph $\mathcal{G}$     (b) Dense representation, $\mathbf{G} := (\mathbf{X}, \mathbf{A})$     (c) Sparse representation, $\mathbf{G} := (\mathbf{V}, \mathbf{E})$, and an $(n, m)$-conditioned part of a Sparse PGC

Figure 2: *An example of a Sparse PGC for a simple graph.* (a) A graph $\mathcal{G}$ with $n = 4$ and $m = 3$. Node types are encoded in shades of green, and edge types in shades of blue. (b) A dense representation of the graph, using a node feature vector $\mathbf{X}$ and an adjacency matrix $\mathbf{A}$. (c) A sparse representation of the same graph, using node features $\mathbf{V}$ and edge features $\mathbf{E}$. Each node vector includes an index (or label), indicated in shades of red, and the type. Each edge vector includes indices of the source and the destination nodes (both in the shades of red), and the edge type. Grey areas represent virtually padded areas that are marginalized out during the computation of the model. Features $\mathbf{V}$ and $\mathbf{E}$ are, first, columnwise flattened, and then concatenated (denoted by the symbol $||$).

sults in an $\mathcal{O}(n^2)$ complexity, even when the graph is sparse and contains much fewer than $n^2$ edges.

Indeed, many real-world graphs are sparse, and, therefore, we exploit this characteristic to propose Sparse PGCs (SPGCs), a new class of tractable generative models that operate directly on a sparse graph representation. A core contribution of our work is the novel idea to model edges explicitly as pairs of node indices, i.e., assigning a probability distribution to them. This representation enables SPGCs to scale with the actual graph size by reducing the complexity to $\mathcal{O}(n + m)$ for graphs with $n$ nodes and $m$ edges. This leads to lower memory usage and faster inference (Figure 1) without compromising exactness or generality. We deploy SPGCs in the context of learning distributions of molecular graphs, demonstrating that they deliver a competitive performance to a wide range of intractable models while offering the capability to perform conditional molecule generation (among other inference queries). We provide the code at `https://github.com/rektomar/SparsePGC`.

## 2 GRAPH REPRESENTATIONS

Throughout this paper, we denote simple (univariate) random variables by uppercase letters, e.g., $X$, and their realizations by corresponding lowercase letters, e.g., $x$. Sets of random variables are denoted by bold uppercase letters, i.e., $\mathbf{X} := \{X_1, \ldots, X_n\}$, with the corresponding realizations denoted by bold lowercase letters, i.e., $\mathbf{x} := \{x_1, \ldots, x_n\}$. We use $[n] := \{1, \ldots, n\}$, where $n \in \mathbb{N}$, to denote a set of positive integers.

**Definition 1** (Graph). We define a graph $\mathcal{G} := (\mathcal{V}, \mathcal{E})$ as a set of vertices, $\mathcal{V} := \{v_1, \ldots, v_n\}$, and a set of edges, $\mathcal{E} := \{(v_i, v_j) \mid v_i, v_j \in \mathcal{V}\}$. We call $N = |\mathcal{V}|$ the number of vertices of $\mathcal{G}$ and $M = |\mathcal{E}|$ the number of edges of $\mathcal{G}$.

We are interested in graphs whose nodes and edges are attributed with categorical random variables. We investigate two possible ways of expressing such graphs: dense

representation (Definition 2) and sparse representation (Definition 3).

**Definition 2** (Dense representation). A dense representation, $\mathbf{G} := (\mathbf{X}, \mathbf{A})$, is a graph (Definition 1) given by the node feature vector, $\mathbf{X} \in \text{dom}(\mathbf{X})$, and the edge adjacency matrix, $\mathbf{A} \in \text{dom}(\mathbf{A})$, where $\text{dom}(\mathbf{X}) := \text{dom}(X)^N$ for $\text{dom}(X) := [n_X]$, and $\text{dom}(\mathbf{A}) := \text{dom}(A)^{N \times N}$ for $\text{dom}(A) := [n_A]$. Here, $n_X$ and $n_A$ are the number of node and edge categories, respectively.

**Definition 3** (Sparse representation). A sparse representation, $\mathbf{G} := (\mathbf{V}, \mathbf{E})$, is a graph (Definition 1) given by the node feature matrix $\mathbf{V} \in \text{dom}(\mathbf{V})$ and the edge feature matrix $\mathbf{E} \in \text{dom}(\mathbf{E})$. The node feature matrix $\mathbf{V} = \{\mathbf{V}_1, \mathbf{V}_2, \ldots, \mathbf{V}_N\}$ is composed of node vectors, $\mathbf{V}_i := (V_i^{\text{idx}}, V_i^{\text{type}})$, where $V_i^{\text{idx}}$ is the index (or label) of $i$-th node and $V_i^{\text{type}}$ is its category, from which it follows that $\text{dom}(\mathbf{V}_i) = [N] \times [n_V]$. Similarly, the edge feature matrix is defined as $\mathbf{E} = \{\mathbf{E}_1, \mathbf{E}_2, \ldots, \mathbf{E}_M\}$, where each edge vector is given by $\mathbf{E}_i := (E_{i,s}^{\text{idx}}, E_{i,d}^{\text{idx}}, E_i^{\text{type}})$. Here, $E_{i,s}^{\text{idx}}$ is the source node index, $E_{i,d}^{\text{idx}}$ is the destination node index, and $E_i^{\text{type}}$ is the edge's category. Consequently, we have $\text{dom}(\mathbf{E}_i) = [N]^2 \times [n_E]$. $n_V$ and $n_E$ are the number of node and edge categories, respectively.

Note that it always holds that $n_E = n_A - 1$, as the dense representation has an additional category to encode the absence of an edge, which further increases its computational complexity.

## 3 SPARSE PGCS

For a graph $\mathbf{g}$ (i.e., a realization of $\mathbf{G}$) with $n$ nodes and $m$ edges, Definition 2 implies that $\mathbf{x}$ contains $n$ values and $\mathbf{a}$ contains $n^2$ values, which results in $\mathcal{O}(n^2 + n)$ size of $\mathbf{g}$. This quadratic complexity is the key disadvantage of the dense representation, as any algorithm relying on it scales poorly to large graphs. In contrast, Definition 3 implies that

**v** contains $2n$ values, this can be reduced to $n$ when node indices are implicit, and **e** containing only $3m$ values, yielding $\mathcal{O}(2n+3m)$ size of **g**. Consequently, the key motivation for our approach is that, for $m \ll n^2$, Definition 3 is particularly advantageous, as it avoids unnecessary overhead of encoding nonexistent edges.

In this section, we define graph circuits for the abstract definition of a graph (Definition 1). Note, however, that this abstract $\mathcal{G}$ can be instantiated by **G** specified in Definition 2 and Definition 3.

**Definition 4** (Graph Scope). Let $\mathcal{G} = (\mathcal{V}, \mathcal{E})$ be a graph (Definition 1). The *scope* of $\mathcal{G}$ is defined as any subset $\mathcal{G}_u = (\mathcal{V}_u, \mathcal{E}_u)$, where $\mathcal{V}_u \subseteq \mathcal{V}$ is an arbitrary subset of nodes, and $\mathcal{E}_u \subseteq \mathcal{E}$ is an arbitrary subset of edges. For graph scopes $\mathcal{G}_a$ and $\mathcal{G}_b$, their union is given by $\mathcal{G}_a \cup \mathcal{G}_b := (\mathcal{V}_a \cup \mathcal{V}_b, \mathcal{E}_a \cup \mathcal{E}_b)$.

**Definition 5** (Graph Circuit). Let $\mathcal{G}$ be a graph (Definition 1). A *graph circuit* (GC) $c$ over $\mathcal{G}$ is a parameterized computational network composed of input, sum, and product units. Each unit $u$ computes $c_u(\mathcal{G}_u)$ based on a graph scope $\mathcal{G}_u$. An input unit is defined as $c_u := f_u(\mathcal{G}_u)$, where $f_u$ is a function over a graph scope $\mathcal{G}_u$. Non-input units $u$ receive the outputs of their input units $\mathsf{in}(u)$ as input. A sum unit is defined as $c_u(\mathcal{G}_u) := \sum_{i \in \mathsf{in}(u)} w_i c_i(\mathcal{G}_i)$, where $w_i \in \mathbb{R}$. A product unit is defined as $c_u(\mathcal{G}_u) := \prod_{i \in \mathsf{in}(u)} c_i(\mathcal{G}_i)$. For sum and product units, $u$, the corresponding graph scope is given by the union of the scopes of their inputs, i.e., $\mathcal{G}_u = \bigcup_{i \in \mathsf{in}(u)} \mathcal{G}_i$.

**Definition 6** (Probabilistic Graph Circuit). A *probabilistic graph circuit* (PGC) over a graph, $\mathcal{G}$, (Definition 1) is a GC (Definition 5) encoding a function, $c(\mathcal{G})$, that is non-negative for all assignments to $\mathcal{G}$, i.e., $\forall_{\mathscr{G}} \in \mathsf{dom}(\mathcal{G}) : c(\mathscr{G}) \geq 0$.

**Definition 7** (Dense PGC). A *dense PGC* is a PGC (Definition 6), where a graph, $\mathcal{G}$, is given in its dense representation, $\mathbf{G} = (\mathbf{X}, \mathbf{A})$ (Definition 2).

**Definition 8** (Sparse PGC). A *sparse PGC* is a PGC (Definition 6), where a graph, $\mathcal{G}$, is given in its sparse representation, $\mathbf{G} = (\mathbf{V}, \mathbf{E})$ (Definition 3).

We choose to instantiate Definition 8 by the following joint probability distribution

$$p(\mathbf{G}) = p(\mathbf{G}^{N,M}, N, M) = p(\mathbf{G}^{n,m}|n,m)\,p(N,M), \quad (1)$$

where $p(\mathbf{G}^{n,m}|n,m)$ is an $(n,m)$-conditioned part of a Sparse PGC over an $n$-node and $m$-edge sparse graph representation, $\mathbf{G}^{n,m}$, and $p(N,M)$ is a joint cardinality distribution characterizing randomness of **G** in the number of nodes, $N$, and the number of edges, $M$.

**Sparse PGCs for simple graphs.** We consider SPGCs for simple graphs, i.e., undirected graphs without self-loops.

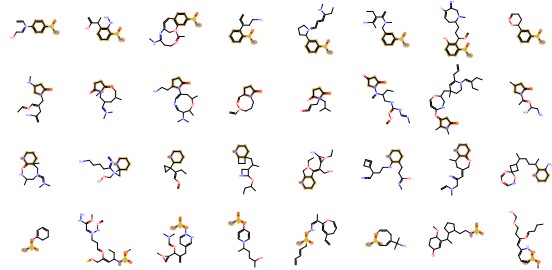

Figure 3: *Conditional generation on the Zinc250k dataset.* The yellow regions highlight the known molecular substructure, which is fixed within each row. Each column displays a new molecule generated conditionally on that substructure.

The topology of a graph $\mathbf{g} = (\mathbf{v}, \mathbf{e})$ is determined by the content of **e**. To reflect the assumption of an undirected and acyclic structure, two structural constraints are imposed on **g**: (i) we neglect the edge direction, i.e., for $\mathbf{e}_i \in \mathbf{e}$, $(e_{i,s}^{\mathsf{idx}}, e_{i,d}^{\mathsf{idx}}, e_i^{\mathsf{type}})$ and $(e_{i,d}^{\mathsf{idx}}, e_{i,s}^{\mathsf{idx}}, e_i^{\mathsf{type}})$ represent the same direction; (ii) we remove all self-loops, i.e., edges of the form $\mathbf{e}_i = (e_{i,s}^{\mathsf{idx}}, e_{i,d}^{\mathsf{idx}}, e_i^{\mathsf{type}})$ for which $e_{i,s}^{\mathsf{idx}} = e_{i,d}^{\mathsf{idx}}$. Furthermore, we instantiate the $(n,m)$-conditioned part of an SPGC as

$$p(\mathbf{G}^{n,m}|n,m) := p(\mathbf{V}^n, \mathbf{E}^m|n,m), \quad (2)$$

where $\mathbf{V}^n$ models the $n$ nodes through their indices and types, and $\mathbf{E}^m$ models the $m$ edges using the indices of the connected nodes and the edge types. To reflect the variability of the input size of $\mathbf{G}^{n,m}$ in our model, we use the marginalization padding [Papež et al., 2025]. This mechanism assumes that a graph can have at most $n_{\max}$ nodes and $m_{\max}$ edges. The actual $n$ nodes and $m$ edges of a graph, $\mathbf{G}^{n,m}$, are then kept intact and the remaining $n_{\max} - n$ unused nodes and $m_{\max} - m$ unused edges are marginalized out. It follows, that the support of the cardinality distribution $p(N, M)$ is $[n_{\max}] \times [m_{\max}]$. Lastly, by making the node indices implicit, i.e., $V_i^{\mathsf{idx}} = i$, they become deterministic variables and can be omitted from the model for simplification. An illustrative example of this setup, including a simple graph, with its dense and sparse representations, as well as the corresponding $(n,m)$-conditioned part of an SPGC, is presented in Figure 2.

**Potential index collision.** Let $\mathbf{g} = (\mathbf{v}, \mathbf{e})$ be a sample drawn from the Sparse PGC. With small but nonzero probability, two forms of index collisions may occur: (i) for some edge $\mathbf{e}_i \in \mathbf{e}$, the source and destination indices may coincide, i.e., $e_{i,s}^{\mathsf{idx}} = e_{i,d}^{\mathsf{idx}}$, resulting in a self-loop; (ii) for edges $\mathbf{e}_i, \mathbf{e}_j \in \mathbf{e}$ with $i \neq j$, it may happen that they encode edge the between the same pair of nodes, i.e., $(e_{i,s}^{\mathsf{idx}}, e_{i,d}^{\mathsf{idx}}) = (e_{j,s}^{\mathsf{idx}}, e_{j,d}^{\mathsf{idx}})$, or $(e_{i,s}^{\mathsf{idx}}, e_{i,d}^{\mathsf{idx}}) = (e_{j,d}^{\mathsf{idx}}, e_{j,s}^{\mathsf{idx}})$. We resolve the two types of collision by sampling without replacement, i.e., we remember which edges have already been sampled, discard the colliding ones, and then sample them again, conditioned on the non-colliding ones.

For more details about the model, see Appendix A and Appendix B, which discuss permutation invariance and

| | QM9 | | | | | Zinc250k | | | | |
|---|---|---|---|---|---|---|---|---|---|---|
| Model | Valid↑ | NSPDK↓ | FCD↓ | Unique↑ | Novel↑ | Valid↑ | NSPDK↓ | FCD↓ | Unique↑ | Novel↑ |
| GraphAF | 74.43±2.55 | 0.021±0.003 | 5.27±0.40 | 88.64±2.37 | 86.59±1.95 | 68.47±0.99 | 0.044±0.005 | 16.02±0.48 | 98.64±0.69 | 100.00±0.00 |
| GraphDF | 93.88±4.76 | 0.064±0.000 | 10.93±0.04 | 98.58±0.25 | 98.54±0.48 | 90.61±4.30 | 0.177±0.001 | 33.55±0.16 | 99.63±0.01 | 99.99±0.01 |
| MoFlow | 91.36±1.23 | 0.017±0.003 | 4.47±0.60 | 98.65±0.57 | 94.72±0.77 | 63.11±5.17 | 0.046±0.002 | 20.93±0.18 | 99.99±0.01 | 100.00±0.00 |
| EDP-GNN | 47.52±3.60 | 0.005±0.001 | 2.68±0.22 | 99.25±0.05 | 86.58±1.85 | 82.97±2.73 | 0.049±0.006 | 16.74±1.30 | 99.79±0.08 | 100.00±0.00 |
| GraphEBM | 8.22±2.24 | 0.030±0.004 | 6.14±0.41 | 97.90±0.14 | 97.01±0.17 | 5.29±3.83 | 0.212±0.005 | 35.47±5.33 | 98.79±0.15 | 100.00±0.00 |
| SPECTRE | 87.30±n/a | 0.163±n/a | 47.96±n/a | 35.70±n/a | 97.28±n/a | 90.20±n/a | 0.109±n/a | 18.44±n/a | 67.05±n/a | 100.00±n/a |
| GDSS | 95.72±1.94 | 0.003±0.000 | 2.90±0.28 | 98.46±0.61 | 86.27±2.29 | 97.01±0.77 | 0.019±0.001 | 14.66±0.68 | 99.64±0.13 | 100.00±0.00 |
| DiGress | 99.00±0.10 | 0.005±n/a | 0.36±n/a | 96.20±n/a | 33.40±n/a | 91.02±n/a | 0.082±n/a | 23.06±n/a | 81.23±n/a | 100.00±n/a |
| GRAPHARM | 90.25±n/a | 0.002±n/a | 1.22±n/a | 95.62±n/a | 70.39±n/a | 88.23±n/a | 0.055±n/a | 16.26±n/a | 99.46±n/a | 100.00±n/a |
| DPGC | 88.83±0.75 | 0.002±0.000 | 1.11±0.01 | 99.38±0.06 | 88.49±0.45 | 14.66±0.66 | 0.043±0.002 | 8.78±0.34 | 100.00±0.00 | 100.00±0.00 |
| SPGC (ours) | 76.21±0.83 | 0.008±0.000 | 1.98±0.04 | 93.90±0.22 | 82.10±0.33 | 23.64±0.91 | 0.118±0.002 | 28.02±0.04 | 99.99±0.02 | 100.00±0.00 |

Table 1: *Unconditional generation on the QM9 and Zinc250k datasets.* We report the mean and standard deviation of molecular metrics for baseline intractable DGMs (top) and tractable DGMs (bottom). The 1st, 2nd, and 3rd best results are highlighted accordingly.

tractability, respectively.

# 4 EXPERIMENTS

We evaluate our approach on the task of molecule generation, where a molecule is represented as a graph, $\mathbf{G}$, with node and edge attributes. Node types correspond to atom types (e.g., C, N, O, etc.), while edge types indicate bond types (SINGLE, DOUBLE, TRIPLE) of a molecule. The main objective is to learn a distribution $p(\mathbf{G})$ over molecular graphs from a dataset, $\{\mathbf{G}_1, \ldots, \mathbf{G}_S\}$, and subsequently generate novel molecules by sampling from the learned distribution.

We conduct three sets of experiments : (1) evaluation of the quality of learned distribution on metrics between the generated samples and samples from the dataset; (2) a comparison of dense (Definition 2) and sparse (Definition 3) graph representations, analyzing memory usage and inference time, with respect to the maximum number of nodes $n_{max}$; and (3) a demonstration of tractability through conditional generation.

**Datasets.** We evaluate a proposed molecule generation task on two benchmark datasets - QM9 [Ramakrishnan et al., 2014] and Zinc250k [Irwin et al., 2012]. To assess computational complexity, we additionally use datasets containing larger molecules, namely Guacamol [Brown et al., 2019] and Polymer [St John et al., 2019]. Summary statistics for all datasets are provided in Appendix C.

**Metrics.** To assess the quality of generated molecules, we use standard molecular generation metrics such as validity, novelty, and uniqueness [Polykovskiy et al., 2020, Brown et al., 2019]. For more detailed comparison, we also deploy a Fréchet ChemNet Distance (FCD) [Preuer et al., 2018] and NSPDK [Costa and Grave, 2010].

**Baselines.** In terms of graph generation, we compare SPGCs with the following intractable DGMs: flow-based models, including GraphAF [Shi et al., 2020], GraphDF [Luo et al., 2021], and MoFlow [Zang and Wang, 2020]; diffusion and score-based models, such as GDSS [Jo et al., 2022], DiGress [Vignac et al., 2023], and EDP-GNN [Niu et al., 2020];

an energy-based model, GraphEBM [Liu et al., 2021]; an autoregressive model, GraphARM [Kong et al., 2023]; and a GAN-based model, SPECTRE [Martinkus et al., 2022]. We also include a tractable model DPGC [Papež et al., 2025] in our comparison of graph generation quality. Additionally, we compare SPGCs and DPGCs in terms of computational complexity.

**Results.** Table 1 presents the performance of SPGCs on the unconditional molecule generation task for the QM9 and Zinc250k datasets. Compared to intractable DGMs, SPGC RT achieves comparable performance in terms of FCD and NSPDK, while also maintaining high validity, uniqueness, and novelty. When compared to the tractable DPGC variants, SPGC RT matches their performance on validity, uniqueness, and novelty, but exhibits slightly weaker results on FCD and NSPDK. In our scalability experiment shown in Figure 1, SPGCs demonstrate superior efficiency: as the maximum number of nodes increases, it consistently consumes less memory and offers faster inference than DPGC. Finally, Figure 3 highlights SPGCs' ability to perform conditional molecule generation, producing diverse and valid samples conditioned on fixed molecular substructures.

# 5 CONCLUSION

We have proposed SPGCs, a tractable DGM for graphs that leverages the sparse graph representation. It models edges using distributions over node indices. This approach fundamentally differs from the previously proposed DPGCs, addressing its scalability limitations while maintaining tractability and having very competitive performance. Our model also achieves highly competitive performance compared to existing intractable DGMs in key metrics, demonstrating its effectiveness in modeling complex distributions over graphs. Additionally, we have provided an illustrative example of superior scalability, highlighting that SPGCs require significantly less memory and offer faster inference times compared to DPGCs. In future work, we aim to further close the performance gap between SPGCs and DPGCs in the FCD and NSPDK metrics. One key limitation we plan to address is the lower validity of tractable models.

**Acknowledgements**

The authors acknowledge the support of the GAČR grant no. GA22-32620S and the OP VVV funded project CZ.02.1.01/0.0/0.0/16_019/0000765 "Research Center for Informatics".

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

# Sparse Probabilistic Graph Circuits
## (Supplementary Material)

**Martin Rektoris**[1]    **Milan Papež**[1]    **Václav Šmídl**[1]    **Tomáš Pevný**[1]

[1]Artificial Intelligence Center, Czech Technical University, Prague, Czech Republic

## A  PERMUTATION INVARIANCE

A permutation of a graph $\mathcal{G}$ with $N$ nodes is a reindexing (or relabelling) of its vertices according to a bijective function $\pi : [N] \to [N]$.[1] Since graphs are a permutation invariant objects, we need to find a way to make $p(\mathcal{G})$ permutation invariant as well, i.e., ensure that $\forall \pi \in \mathbb{S}^n : p(\pi\mathcal{G}) = p(\mathcal{G})$, where $\mathbb{S}_n$ is a set of all permutations of $[n]$. For a graph in the sparse representation, $\mathbf{G} = (\mathbf{V}, \mathbf{E})$, we define its permutation as $\pi\mathbf{G} := (\pi\mathbf{V}, \pi\mathbf{E})$, where $\pi\mathbf{V} := \{\pi\mathbf{V}_{\pi(1)}, \dots, \pi\mathbf{V}_{\pi(N)}\}$, with $\pi\mathbf{V}_i := (\pi(V_i^{\text{idx}}), V_i^{\text{type}})$, and $\pi\mathbf{E} := \{\pi\mathbf{E}_{\pi_E(1)}, \dots, \pi\mathbf{E}_{\pi_E(M)}\}$, with $\pi\mathbf{E}_i := (\pi(E_{i,s}^{\text{idx}}), \pi(E_{i,d}^{\text{idx}}), E_i^{\text{type}})$. Here, $\pi_E : [M] \to [M]$ is a pemutation of the edge indices, defined as $\pi_E := h(\pi, \mathbf{E})$, where $h$ is a deterministic function. One possible implementation of $h$ proceeds as follows:

1. Construct the adjacency matrix $\mathbf{A}$ from $\mathbf{E}$.

2. Apply the permutation $\pi$ to $\mathbf{A}$ to obtain the permuted adjacency matrix $\pi\mathbf{A}$, where $(\pi\mathbf{A})_{ij} := \mathbf{A}_{\pi(i)\pi(j)}$ for $i, j \in [N]$.

3. Extract non-zero entries of the lower triangle of $\pi\mathbf{A}$ in a given traversal order, e.g., the row-major order, to obtain the permuted edge list $\pi\mathbf{E}$.

4. Compute $\pi_E$ as the permutation that aligns the original edges $\mathbf{E}$ with the new edges $\pi\mathbf{E}$.

An example of this implementation is shown in Figure 4. Note that the specific behavior of $h$ in step 3 may vary depending on the traversal strategy (e.g., the row-major order vs. the column-major order), which then influences the resulting $\pi_E$.

To ensure permutation invariance of $p(\mathbf{G}^{n,m}|n, m)$, we sort each input graph $\mathbf{G}$ into its canonical order and before computing its likelihood:

$$p(\mathbf{G}^{n,m}|n, m) := p(\text{sort}(\mathbf{V}^n, \mathbf{E}^m)|n, m). \tag{3}$$

The canonical order is determined by the graph's permutation $\pi_c := \pi_c(\mathbf{G}^{n,m})$, computed based on the graph's realization. As discussed in [Papež et al., 2025], this procedure leads to a lower bound on the likelihood $p(\mathbf{V}^n, \mathbf{E}^m|n, m) \geq p(\text{sort}(\mathbf{V}^n, \mathbf{E}^m)|n, m) = p(\mathbf{V}^n, \mathbf{E}^m|\pi_c, n, m)$. In our experiments, we use the canonicalization algorithm for molecular graphs from RDKit [Landrum et al., 2006] and the associated definition of $h$ it implicitly induces.

## B  TRACTABILITY

DPGCs [Papež et al., 2025] discuss several approaches to achieve permutation invariance. All these approaches involve trade-offs that in either the model loses strict tractability by requiring evaluation over all $n!$ node orderings, or it sacrifices expressivity. In our work, we choose sorting, which introduces a lower bound on the likelihood (as discussed in Appendix A). As a result, strict tractability, as formally defined by Papež et al. [2025], becomes out of reach. However, this choice enables computational feasibility by evaluating only a single canonical order instead of all permutations. Despite this relaxation, the

---

[1]We use $\pi$ in two different settings. The first one, denoted as $\pi(\cdot)$, where the input is from the domain $[N]$, uses parentheses around the input. For the second one, $\pi\cdot$, where the input is a complex object, e.g., a set or a vector, no parentheses are used. In the case we use the second one, we further define its meaning in the text if necessary.

*Accepted for the 8th Workshop on Tractable Probabilistic Modeling at UAI  (TPM 2025).*

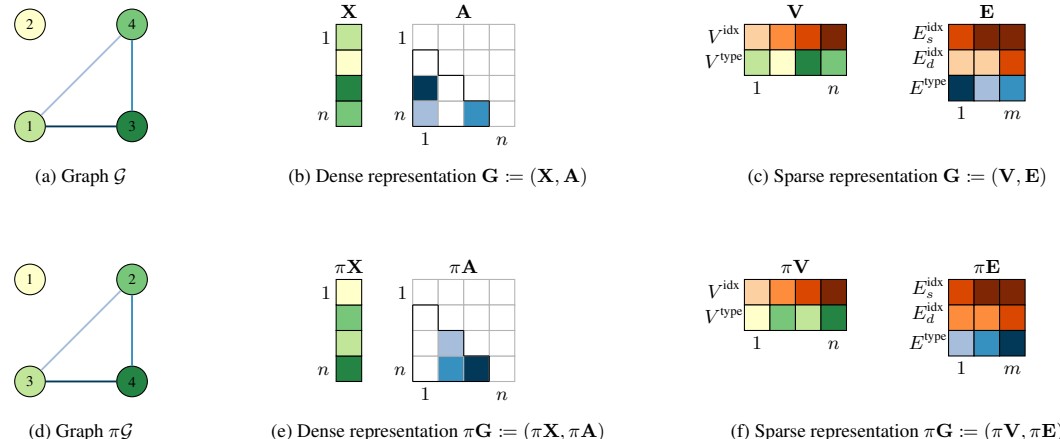

(a) Graph $\mathcal{G}$     (b) Dense representation $\mathbf{G} := (\mathbf{X}, \mathbf{A})$     (c) Sparse representation $\mathbf{G} := (\mathbf{V}, \mathbf{E})$

(d) Graph $\pi\mathcal{G}$     (e) Dense representation $\pi\mathbf{G} := (\pi\mathbf{X}, \pi\mathbf{A})$     (f) Sparse representation $\pi\mathbf{G} := (\pi\mathbf{V}, \pi\mathbf{E})$

Figure 4: *An example of a permutation $\pi$ applied to a graph $\mathbf{G}$. In this example, we use the node permutation $\pi(1) = 3, \pi(2) = 1, \pi(3) = 4, \pi(4) = 2$, which induces the following edge permutation: $\pi_E(1) = 2, \pi_E(2) = 3, \pi_E(3) = 1$, by traversing $\mathbf{A}$ and $\pi\mathbf{A}$ row-wise.*

model retains its algebraic tractability, which allows us to compute analytical integration. In this sense, we establish the tractability of SPGCs by adopting Proposition 1 and Definition 6 from [Papež et al., 2025].

**Proposition 1** (Tractability of Sparse PGCs). *Let $p$ be an SPGC such that $p(\mathbf{G}^{n,m}|n, m)$ is tractably $\mathbb{S}_n$-invariant, and $p(N, M)$ has a finite support. Furthermore, consider that $\mathbf{G}$ can be decomposed into two subgraphs $\mathbf{G} = (\mathbf{G}_a, \mathbf{G}_b)$, where $\mathbf{G}_a$ has a random size and $\mathbf{G}_b$ has exactly $k$ nodes and $l$ edges. Then $p(\mathbf{G})$ is tractable if there exists $d \in \mathbb{N}$ such that*

$$\int p(\mathbf{g}_a, \mathbf{G}_b)d\mathbf{g}_a = \sum_{n=k}^{\infty} \sum_{m=l}^{\infty} \int p(\mathbf{g}_a^{n-k,m-l}, \mathbf{G}_b^{k,l})p(n, m)d\mathbf{g}_a^{n-k,m-l} \tag{4}$$

*can be computed exactly in $\mathcal{O}(|p|^d)$ time.*

Since $p(N, M)$ has a finite support, the two infinite sums above reduce to finite ones. Computing this integral is crucial to a broad range of probabilistic queries.

## C    EXPERIMENTAL DETAILS

All experiments were run on Nvidia Tesla V100 GPUs with a 4-hour time limit per job, managed via the SLURM job scheduler. We used an 80/10/10 train/validation/test split ratio and evaluated each model over five runs with different random initializations. Models were trained by maximizing log-likelihood using the Adam optimizer [Kingma and Ba, 2014] for 40 epochs with a learning rate of 0.05, decay rates $(\beta_1, \beta_2) = (0.9, 0.82)$ and a batch size of 256. After training, we unconditionally sampled $10,000$ molecular graphs to compute the molecular metrics. Finally, we report the mean and standard deviation of the metrics across the runs, based on the model selected from the grid search (using hyperparameters from Table 3) that achieved the highest validity. The best validity model was also used to generate figures with both conditional and unconditional molecules. For scalability analysis shown in Figure 1, we used the hyperparameters of the best validity model on the Zinc250k dataset; additionally, we recovered the best hyperparameters for DPGCs from their reported settings for comparison.

| Dataset | $|\mathcal{D}|$ | $n_{\max}$ | $m_{\max}$ | $n_V$ | $n_E$ |
|---|---|---|---|---|---|
| QM9 | 133,885 | 9 | 12 | 4 | 3 |
| Zinc250k | 249,455 | 38 | 45 | 9 | 3 |
| Guacamol | 1,273,104 | 88 | 87 | 12 | 3 |
| Polymer | 76,353 | 122 | 145 | 7 | 3 |

Table 2: *Summary statistics of the molecular datasets.* $|\mathcal{D}|$ denotes the number of instances in each dataset, $n_{\max}$ the maximum number of nodes, $m_{\max}$ the maximum number of edges, $n_V$ the number of node categories, and $n_E$ the number of edge categories.

| Dataset | PC | | $n_L$ | $n_S$ | $n_I$ | $n_R$ | $n_c$ |
|---|---|---|---|---|---|---|---|
| QM9 | BT | $V^{\text{type}}$ | $\{2,3\}$ | $\{16,32\}$ | $\{16,32\}$ | - | $\{256\}$ |
| | | $E^{\text{idxs}}$ | $\{3,4\}$ | $\{16,32\}$ | $\{16,32\}$ | - | |
| | | $E^{\text{type}}$ | $\{2,3\}$ | $\{16,32\}$ | $\{16,32\}$ | - | |
| | RT | $V^{\text{type}}$ | $\{2,3\}$ | $\{16,32\}$ | $\{16,32\}$ | $\{8,16\}$ | $\{256\}$ |
| | | $E^{\text{idxs}}$ | $\{3,4\}$ | $\{16,32\}$ | $\{16,32\}$ | $\{8,16\}$ | |
| | | $E^{\text{type}}$ | $\{2,3\}$ | $\{16,32\}$ | $\{16,32\}$ | $\{8,16\}$ | |
| Zinc250k | BT | $V^{\text{type}}$ | $\{4,5\}$ | $\{16,32\}$ | $\{16,32\}$ | - | $\{256\}$ |
| | | $E^{\text{idxs}}$ | $\{5,6\}$ | $\{16,32\}$ | $\{16,32\}$ | - | |
| | | $E^{\text{type}}$ | $\{4,5\}$ | $\{16,32\}$ | $\{16,32\}$ | - | |
| | RT | $V^{\text{type}}$ | $\{4,5\}$ | $\{16,32\}$ | $\{16,32\}$ | $\{8,16\}$ | $\{256\}$ |
| | | $E^{\text{idxs}}$ | $\{5,6\}$ | $\{16,32\}$ | $\{16,32\}$ | $\{8,16\}$ | |
| | | $E^{\text{type}}$ | $\{4,5\}$ | $\{16,32\}$ | $\{16,32\}$ | $\{8,16\}$ | |

Table 3: *Hyperparameters for different variants of SPGCs.* PC denotes the type of region graph [Dennis and Ventura, 2012] used by the probabilistic circuit: BT stands for binary tree, and RT for random binary tree [Peharz et al., 2020b]. $n_L$ is the number of PC layers, $n_S$ is the number of children per sum node, $n_I$ is the number of input units per variable, $n_R$ is the number of repetitions for RT-based PCs, and $n_C$ is the number of children of the top-level sum node. The model input layer is a categorical distribution; however, the $V^{\text{type}}$, $E^{\text{idxs}}$, and $E^{\text{type}}$ parts of the input each have a different number of categories. Our implementation accounts for this by allowing distinct hyperparameters for each part of the input.

# D   ADDITIONAL RESULTS

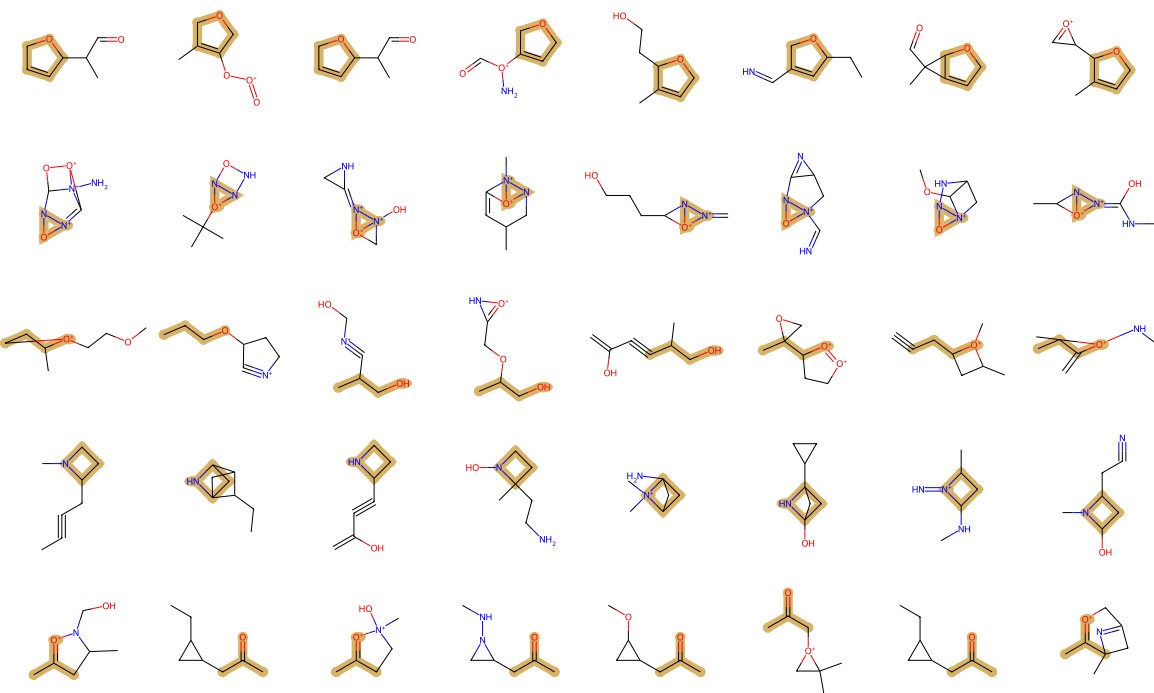

Figure 5: *Conditional generation on the QM9 dataset.* The yellow regions highlight the known molecular substructure, which is fixed within each row. Each column displays a new molecule generated conditionally on that substructure.

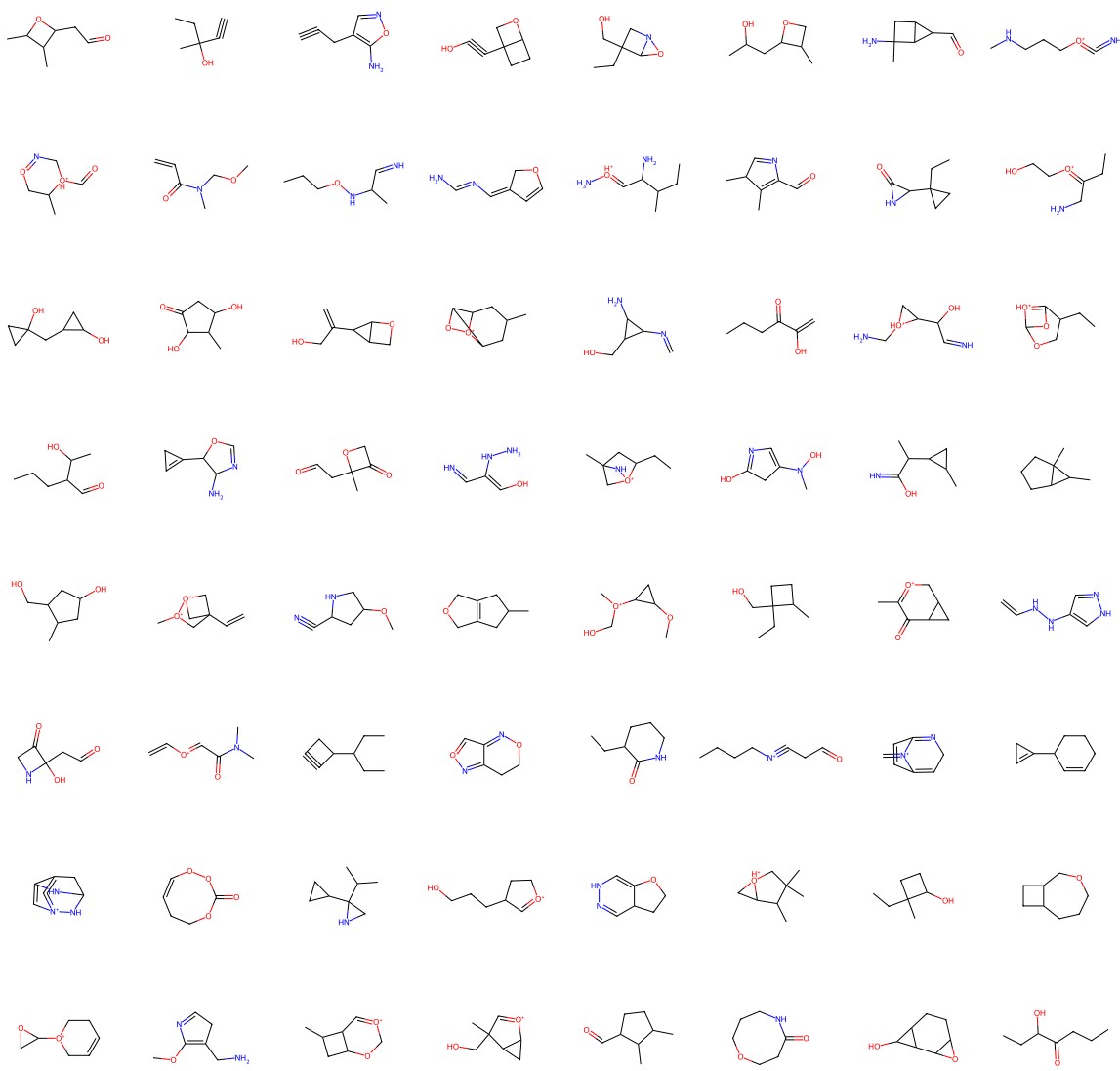

Figure 6: *Unconditional generation on the QM9 dataset.*

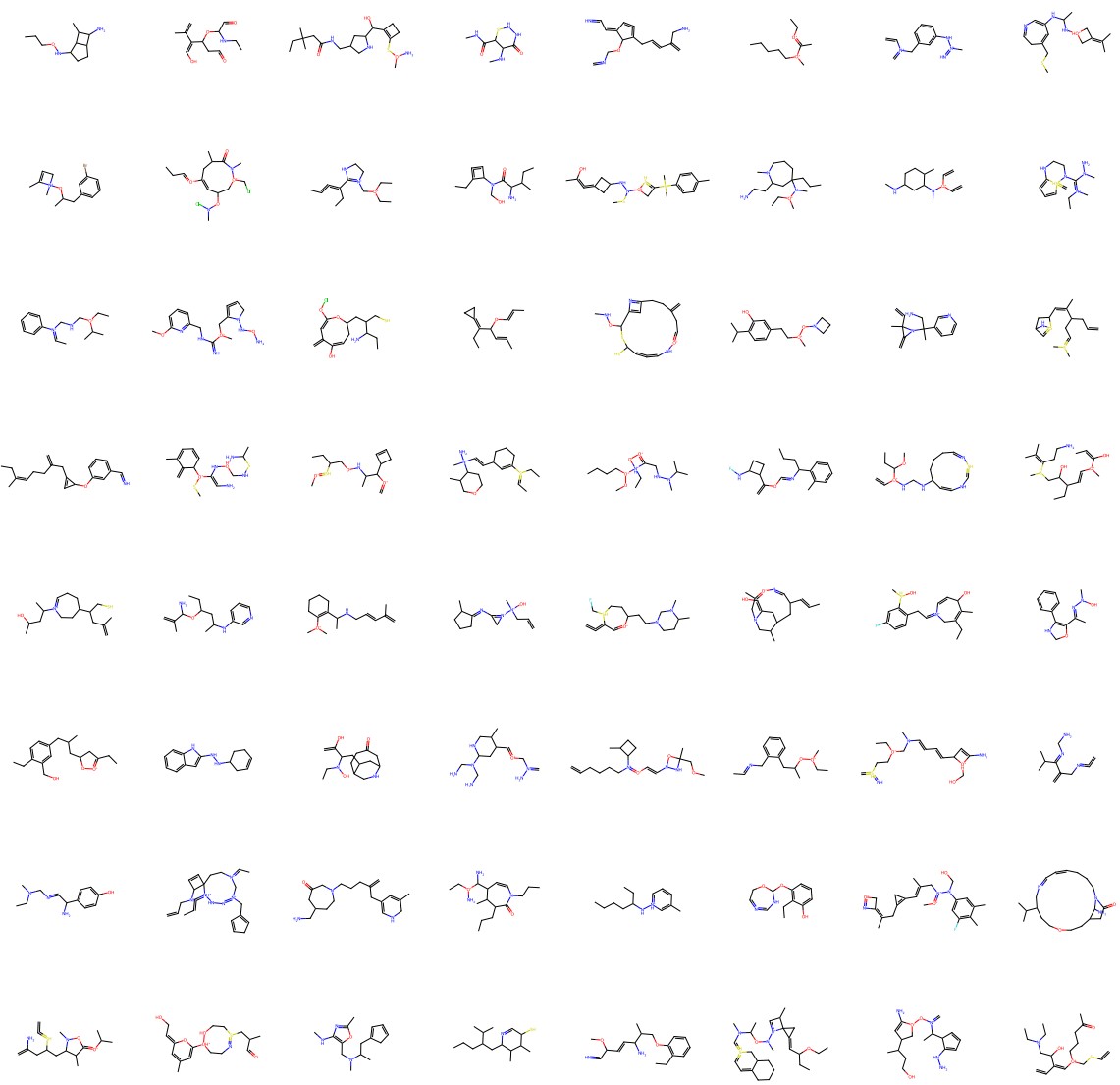

Figure 7: *Unconditional generation on the Zinc250k dataset.*