# OpenReview forum: "Sparse Probabilistic Graph Circuits"
_auai.org/UAI/2025/Workshop/TPM — TPM 2025_

### Official Review · Reviewer_177g · 2025-06-07
**Review for Sparse Probabilistic Graph Circuits**

**Rating:** 3

**Review:**

Strengths

1. Relevant and timely topic: The paper addresses an important computational problem in the field of AI—specifically tractable inference over graph-structured data. The proposed method fits well within the broader effort to handle uncertainty in deep models, making it a strong and appropriate submission for the workshop.

2. Strong computational results: The paper demonstrates a significant improvement in inference efficiency, reducing the complexity from quadratic time O(N²) to a more scalable linear form O(N+M). This is a notable technical advancement, and the speed-up has meaningful implications for real-world applications that rely on fast sampling or marginal inference over large graphs.

Weaknesses

1. Insufficient detail on datasets: The paper lacks depth in its discussion of the dataset characteristics, including size, quality, and preprocessing steps. More clarity on the nature of the dataset would help the reader assess the validity and generality of the experimental results.

2. Unclear presentation of new definitions: Some of the newly introduced definitions and terms are not standard in the literature. While they are important to the method, their inclusion in the main paper adds unnecessary complexity. It would be more effective to move these to the supplementary material and instead use the space to expand on experimental design and analysis.

---

### Official Review · Reviewer_jozo · 2025-06-12
**A simple yet surprisingly effective way to scale probabilistic circuits on large graph distributions**

**Rating:** 3

**Review:**

The paper introduces a variant of Probabilistic Graph Circuits (PGCs) employing a memory efficient sparse formalization of vertex and edge variables. These models, called sparse PGCs, are shown to speed-up inference and require less memory w.r.t. PGCs. At the same time, sparse PGCs achieve performances on unconditional generation of molecules that are competitive with dense PGCs.

The paper is extremely well written, with a well thought structure and nice figures. The TPM workshop is definitely the right venue for this short paper, so I recommend acceptance. I also believe there is plenty of opportunities in making it a full paper.

One thing that I believe is missing is showing how performances change when increasing the bound on the number of edges $m_{\text{max}}$. In fact, one could ideally recover the same performances of dense PGCs by making $m_{\text{max}}$ large enough, but in practice this might not be the case. It is also not clear how efficient these sparse PGCs are when compared to other generative model listed in Table 1. Furthermore, I recommend the authors to put an example of the circuit structure being used (i..e, the PC block in Figure 2).

The authors mention potential index collisions at the end of page 3. Although sampling with replacement might be sufficient in solving these problems, I have the feeling that many variables will eventually need to be re-sampled, especially as one increases $m_{\text{max}}$. It might be possible to encode logical constraints ensuring that index collisions never occur using a logical circuit [A] [B].

[A] Ahmed et al. Semantic Probabilistic Layers for Neuro-Symbolic Learning (2022).
[B] Loconte et al. How to Turn Your Knowledge Graph Embeddings into Generative Models (2023).

**Nominate For Best Paper:**

["Yes"]